# An Overview on Total Valorization of *Litsea cubeba* as a New Woody Oil Plant Resource toward a Zero-Waste Biorefinery

**DOI:** 10.3390/molecules26133948

**Published:** 2021-06-28

**Authors:** Yufei Qiu, Yasi Yu, Ping Lan, Yong Wang, Ying Li

**Affiliations:** 1Guangdong International Joint Research Center for Oilseeds Biorefinery, Nutrition and Safety, Department of Food Science and Engineering, Jinan University, Guangzhou 510632, China; qiuyufei224@stu2020.jnu.edu.cn (Y.Q.); yuyasi99@stu2017.jnu.edu.cn (Y.Y.); 2Faculty of Pharmacy, Institute for Advanced and Applied Chemical Synthesis, Jinan University, Guangzhou 510632, China; ping.lan@jnu.edu.cn; 3Qingyuan Yaokang Biotechnology, Qingyuan 513200, China

**Keywords:** woody oil plant resource, *Litsea cubeba* valorization, essential oil bioactivities, citral-based value-added products, kernel oil and protein, zero-waste sustainable biorefinery

## Abstract

With the increasing global demand for edible oils and the restriction of arable land minimum in China, woody oil plants have gradually become the optimal solution to cover the shortage of current edible oil supply and to further improve the self-sufficiency rate. However, due to the lack of knowledge and technique, problems like “how to make full use of these plant resources?” and “how to guide consumers with reasonable data?” limit the development of woody oilseed industry towards a sustainable circular economy. In this review, several emerging unique woody oil plants in China were introduced, among which *Litsea cubeba* as a new woody oil plant was highlighted as a reference case based on its current research progress. Unlike other woody oil plants, essential oil rather than oil from *Litsea cubeba* has always been the main product through the years due to its interesting biological activities. Most importantly, its major component, citral, could be the base for other synthesized perfume compounds with added value. Moreover, the sustainable biorefinery of large amounts of waste residual after *Litsea cubeba* essential oil processing is now technically feasible, which could inspire a total valorization pathway for other woody oil plants to make more competitive plant-based products with both economic, social, and ecological benefits.

## 1. Introduction

Woody oil plant is the general term for the fruits or seeds of woody plants that can be used for oil extraction. It is a renewable resource for a diversity of oil products, such as edible oils, aromatic oils, industrial oils, and even oils for bioenergy [1]. The exploitation of woody oil plant based on the biorefinery concept is superior to conventional agriculture and more adaptable to climate change, which has advantages of using marginal land resources to develop value-added products and alleviating economic growth from dependence on fossil energy to some extent. Moreover, the industrial development of woody oil plants is also conducive to reduce the tight supply of edible oil, and meanwhile, maintain both economic stability and ecological security.

It is stated by the General Office of the State Council of China that 800 key counties would plant woody oilseeds like camellia, walnut, and peony by 2020, and a batch of standardized and industrialized demonstration bases would be established, resulting in the growing planting area from the existing 8 million hectares to around 13 million hectares, with an annual output of about 1.5 million tons of woody edible oils. Although woody oil plants in China are diverse and abundant, they are still unexploited or underdeveloped due to lack of scientific knowledge and systematic technical studies on intensive processing and integrated utilization.

The depletion of fossil energy and the unprecedented global climate change force people to eagerly seek raw materials for the available biomass, responding to the demand of sustainable energy supply [2,3]. In recent years, with the rapid development of society and the increasing demand for a better life, some local woody oil plants in China like *Camellia oleifera* have gradually attracted more and more attention in both domestic and foreign markets due to their unique chemical compositions and functional properties [4]. Besides this, *Litsea cubeba* was found as an emerging woody oil plant resource that can be implemented in a total valorization toward a zero-waste biorefinery, resulting in various plant-based products with added value for different application fields.

## 2. Woody Oil Plant Resources in China

As a big agricultural country with vast territory and excellent natural conditions, China has a wide variety of indigenous woody oil plants such as *Camellia oleifera*, *Juglans regia*, *Paeonia suffruticosa*, *Xanthoceras sorbifolium*, *Acer truncatum*, *Eucommia ulmoides*, and *Swida wilsoniana*, whose morphological properties, seed oil content, and characteristic bioactive ingredients are shown in Table 1.

### 2.1. Camellia oleifera

*Camellia oleifera* is one of four major woody oil plants in the world, and its seed oil enjoys the reputation of “Oriental olive oil” because of its similar fatty acid composition to that of olive oil in Europe. Camellia oil is rich in various bioactive ingredients, and long-term intake of camellia oil can regulate immune function; decrease cholesterol levels; lower blood sugar and lipids; and prevent hypertension, hepatic, cardiovascular, and neurological diseases [21,22,23]. Moreover, it has strong antioxidant activity and high thermal stability after refining; its good affinity to human skin can help to improve skin conditions and even reduce skin aging [24,25,26]. Therefore, it can be used as a cosmetic oil to make natural skin care products. In addition, camellia shell contains a large amount of cellulose, lignin, and hemicellulose, which can be used to produce industrial raw materials such as furfural, xylitol, tannin, and even biochar that can effectively remove fluoride in water [27].

### 2.2. Juglans regia 

*Juglans regia* is one of the important nuts and woody oil species in China, where the planting area and total output of walnut rank first in the world. The oil content of walnut is almost the highest among all woody oils, which is known as “oil depot on the tree”. Since walnut oil is rich in a variety of biologically active substances, they have important nutritional and health care values such as the function of strengthening brain and body, moistening lungs, and even beautifying skin. Besides, the integrated utilization of by-products from walnut oil processing has gradually become a hotspot, which greatly improves the economic benefits of walnuts. For instance, walnut pomace, husk, and oil cakes can be further processed to produce soy sauce, activated carbon, milk, pigment, polysaccharides, polyphenols, protein powder, and peptides [28,29,30,31,32].

### 2.3. Paeonia suffruticosa

*Paeonia suffruticosa* is an emerging woody oil plant with its unique flowers signifying “Celestial Beauty, the king of flowers”, which is native to the basin of Yangtze River and Yellow River in China. The oil content and quality in peony seed is high, it has a comparable content of α-linolenic acid to that in flaxseed oil, which has been reported to have the effect of lowering blood fat and improving cardiovascular disease and brain nerve function, as well as being anti-cancer and anti-inflammatory [33,34,35]. Furthermore, there are also many concomitant minor nutrients in peony seed oil, which have broad application prospects in the fields of medicine, cosmetics, and functional food.

### 2.4. Xanthoceras sorbifolium

*Xanthoceras sorbifolium* is a rare edible oil tree species, which is native to the Loess Plateau in northern China. It has now become the first selection species in desertification control and environmental protection. *Xanthoceras sorbifolium* seed oil is not only suitable to edible uses, but also can be applied in bioenergy field due to its similar chemical composition to that of ideal diesel substitutes. The quality indicators of biodiesel prepared from *Xanthoceras sorbifolium* oil as the feedstock met the condition and relevant regulations of fuel substitutes [36,37]. In addition, *Xanthoceras sorbifolium* oil also contains a variety of mineral elements and active ingredients, which has the potential to be applied in the field of medical care, functional food, and cosmetics.

### 2.5. Acer truncatum

*Acer truncatum* is another unique tree species in China, which was named after its samara shaping, like gold ingot from ancient China. Its seed oil was also approved as a new edible oil resource by the National Health Commission in China. Unlike other oils, *Acer truncatum* seed oil has a high content of unsaturated fatty acid (≥90%), essential fatty acids (>50%), and multivitamins (e.g., vitamin E content is the highest among all plant oils), which brings itself great development prospects and economic benefits. The plant containing nervonic acid is scarce worldwide, thus *Acer truncatum* seed oil containing around 5.5% of nervonic acid can be refined purposively and applied in medical drugs, health foods, and other fields, especially for the prevention and treatment of major neurological diseases such as Alzheimer’s disease and Parkinson’s disease [38,39].

### 2.6. Eucommia ulmoide

*Eucommia ulmoides* is a unique and precious woody oil plant in China, and its leaves, flowers, and fruits have medicinal and edible values. Hence, it is known as “Plants gold”. *Eucommia ulmoides* seed oil is a kind of high-grade plant oil, which has remarkable pharmacological functions such as reducing blood fat and improving cardiovascular disease. It is interesting that *Eucommia ulmoides* gum extracted from its leaves has the durability of rubber and plastic, though it has no genetic relationship with rubber tree, which can be processed into polymer materials with different properties [40]. Such gum can also partially replace natural rubber and blend with other synthetic rubbers or plastics, which can expand its application in transportation, communication, and construction fields [41,42,43].

### 2.7. Swida wilsoniana

*Swida wilsoniana* is cultivated in China as an important woody oil plant with wide ecological adaptability. Both its pulp and kernels contain lots of oil with high value in both food and non-food field [18,44], in which the main fatty acid compositions are linoleic acid and oleic acid. *Swida wilsoniana* oil was approved as a new resource for food in 2013. Its oil is rich in unsaturated fatty acids, vitamin E, β-carotene, and other functional ingredients, which not only have hypolipidemic function, but also help to supply essential fatty acids and prevent fatty liver and atherosclerosis [45,46]. In addition, studies have shown that the *Swida wilsoniana* oil could be catalyzed to produce high-quality biofuels, which can be used as an alternative energy source with less competition with foods [47,48].

### 2.8. Litsea cubeba

*Litsea cubeba* is a special woody oil plant resource in China, which is mainly distributed in provinces and regions in the south of the Yangtze River. For the past several decades, *Litsea cubeba* essential oil has been the only product for trading, which can be extracted from its whole plant, including leaves, flowers, fruits, trunks, and even roots. However, the limited essential oil yield (≈4%) results in a large amount of kernel and pomace as wastes, which are underutilized and usually discarded to cause environmental burden. China is the largest producer and exporter o *Litsea cubeba* essential oil in the world. Most European countries import essential oils for further purification to obtain high-value functional products, and *Litsea cubeba* essential oil is no exception. Its main component citral has been used for the synthesis of high-grade fragrances (e.g., ionone, geraniol, citronellol and irone), and vitamin A and E as well. Apart from this, very few value-added processing studies have been reported for other residues after *Litsea cubeba* essential oil distillation. Considering this, this review proposes a total valorization pathway of *Litsea cubeba* toward a zero-waste biorefinery based on its current development status (Figure 1), which helps to make full use of this new woody oil plant resource to have various bio-based products with greater value (e.g., essential oil, citral-based derivatives, kernel oil, fatty acid, biofuel, surfactants, fodder, etc.) in different fields. The integrated utilization of *Litsea cubeba* may inspire a sustainable biorefinery route to produce plant-based products from other woody oil plant resources, resulting in the optimal compromise solution between economy, society, and environment.

## 3. *Litsea cubeba* Essential Oil and Citral-Based Derivatives

*Litsea cubeba* is a deciduous shrub or a small tree belonging to the Lauraceae family, which is a characteristic potential industrial crop in China. This plant is photophilous and shade tolerant, the good germination of its seed can make it easy to survive in the undergrowth or at the roadside. The yield and composition of essential oils vary depending on plant organs extracted (i.e., fruits, leaves, flowers, roots, and trunk), environmental conditions (e.g., soil, altitude, climate and irrigation), and other parameters (e.g., harvest time, ripeness and postharvest technologies). The main chemical constituents of *Litsea cubeba* essential oil are terpenoids, especially for citral, whose biological activities including antioxidant, anti-inflammatory, antimicrobial, and anthelmintic capacities, have attracted more interests in recent years. Furthermore, high value-added synthetic products derived from citral have also been favored by enterprises in the field of perfume, flavor and fragrance, and nutraceuticals.

### 3.1. Extraction of Litsea cubeba Essential Oils

It is reported that the yield of essential oil from *Litsea cubeba* fruits is higher than that from other plant organs [49]. The traditional methods for extracting essential oil from *Litsea cubeba* is hydrodistillation, which is simple to operate but usually with a low extraction rate. Moreover, the long treatment time may cause unexpected changes in chemical composition and smell. Hence, innovative extraction technologies (e.g., microwave-assisted extraction, ultrasonic-assisted extraction, enzymatic-assisted extraction, supercritical fluid extraction, and other combined extractions) have been developed for essential oils in the past decade [50,51,52,53,54], which can not only enhance the extraction efficiency in a shorter time, but also maintain or improve the quality of essential oils.

As summarized in Table 2, emerging techniques like microwave and ultrasound could effectively enhance the extraction of *Litsea cubeba* essential oils with good quality, which paved the way for the total valorization of this plant. The strong penetrating power of electromagnetic microwaves could accelerate diffusion and transfer of essential oils from interior gland cells of fruit and peel tissues to exterior. Compared to traditional steam distillation, the optimized microwave-assisted extraction (MAE) could increase the yield by 36.5–37.5% but shorten the treatment time by four times [55]. Moreover, the citral content in essential oils was 5% higher and the amount of its loss in purification reduced by 33.3%. MAE was also optimized with an average essential oil yield of up to 10.29% (g/g), which could improve to 14.19% (g/g) with the combination of ultrasound [56].

Ultrasonic cavitation can speed up the breakage of plant cell walls, which is conducive to the release of essential oils and the penetration of solvents. Ultrasonic-assisted vacuum extraction was reported to have an extraction rate of 6.94% under optimal conditions, which had a citral content of as high as 87.65% [57]. Peng et al. [58] found that the yield of essential oils could be enhanced with the assistance of ultrasound, where the yield reached the highest when ultrasonic time was 25 min. Although crushing could increase the extraction ratio, the purity of essential oils was lower, which was not recommended to the extraction.

It is well known that enzymes are highly specific and have high catalytic efficiency with mild reaction conditions. Generally, the use of appropriate enzymes can improve the extraction rate without any damages to the raw materials, which is beneficial to the further exploitation of the residues after extraction. The yield of essential oil varies widely depending on relevant factors such as enzyme types and dosages. Xie et al. [59] used recombinant expansin and cellulase to extract the *Litsea cubeba* essential oil, which significantly increased the yield of essential oil compared to conventional extraction. The function of expansin is to break the hydrogen bond between cellulose microfilaments and hemicellulose. The interaction of expansin and hydrolase facilitates expansion and disruption of plant cell wall without any losses of nutrients. Therefore, expansin, as a product of genetic engineering, is worthy to be further explored regarding the high price and large amount of enzyme used in the large-scale extraction.

The type of solvent is of great importance to extraction. Supercritical CO_2_ is one recognized green solvent for the extraction of bioactive components from natural plants, considering its excellent extractive properties, safety, eco-friendliness, and economica viability [62]. Supercritical CO_2_ extraction could not only avoid oxidation, degradation, and organic solvent residues resulting from solvent extraction and distillation, but also overcome the disadvantages of pressing methods like the volatilization of essential oils and color darkening. Zhang et al. [60] studied the influencing factors in the supercritical CO_2_ extraction of *Litsea cubeba* essential oil, where the particle size, CO_2_ flow rate, extraction pressure, temperature, and time were optimized to obtain the highest extraction ratio. Although supercritical CO_2_ extracts could maintain the naturalness of the raw materials, there are still some limitations for wide industrial applications of this technique like poor robustness of the early commercial equipment, lack of standard extraction procedures, difficulties in extracting polar compounds, and inefficiency in clean-up [63].

### 3.2. Bioactivities of Litsea cubeba Essential Oils

In China, *Litsea cubeba* is also called ‘Bì Chéng Qié’, which has been commonly used as a traditional herbal medicine. From the perspective of traditional Chinese medicine, fruits, root bark, and leaves from *Litsea cubeba* can be used as medicines to treat stomach pain, vomit, fever, snake bites, swelling, cholera, and detoxification. Modern pharmacology studies have shown that *Litsea cubeba* extracts have a variety of biological activities such as antioxidant, antimicrobial, anti-inflammatory, anti-tumor, and insecticide, which gives it the potential to be widely used in the field of agriculture, medicine, food, flavor, and fragrance, etc.

#### 3.2.1. Antimicrobial Activity

Citral in *Litsea cubeba* essential oil was firstly found to have a strong inhibitory effect on bacteria [64]. Many experiments have shown that *Litsea cubeba* essential oil had a broad-spectrum bacteriostasis and a certain antifungal effect. Minimal inhibitory concentrations (MICs) of *Litsea cubeba* essential oil against different bacteria and fungi are summarized in Table 3, which shows a great potential for wide applications ranging from grain storage to preservation of fruit, aquatic and cosmetic products, etc.

You et al. [78] found that there were many endophytic bacteria in *Litsea cubeba* plants. Such bacterial population was also significant for the increase of *Litsea cubeba* essential oil yield and citral content in oil, which could enhance its bacteriostatic effect. Wang et al. [75] studied the bacteriostatic effect of essential oils from three species of May Chang tree, among which the content of essential oil from *Litsea cubeba* (Lour.) Pers. was significantly higher than those from *Litsea mollis* Hemsl. and *Litsea cubeba* var. *formosana*. Moreover, its bacteriostatic effect on *Fusarium oxysporum* f. sp. *fordii* 1, *Escherichia coli*, and *Listeria monocytogenes* was also better than the other two Litsea essential oils, which could be potentially developed for antibacterial and antiseptic products. In addition, the contents of citral (70–90%) and linalool were shown to have a significant positive correlation with inhibition rate.At present, studies about bacteriostatic mechanism of *Litsea cubeba* essential oils mainly focus on the destruction of cell’s biological structure and the inhibition of their normal life activities. *Litsea cubeba* essential oil could destroy the cell membrane of methicillin-resistant *Staphylococcus aureus* (MRAS), resulting in the loss of biological macromolecules in cells. Moreover, it could inhibit the respiratory metabolism of MRSA through reducing the activity of glucose-6-phosphate dehydrogenase, a key regulatory enzyme in hexose monophosphate pathway (HMP). Besides, the main component citral could also form a chimera with DNA of MRSA to inhibit the biological activity of bacteria [66]. Similarly, the antibacterial mechanism of *Litsea cubeba* essential oil on Enterohemorrhagic *Escherichia coli* O157:H7 was reported to be related to increasing the permeability of cell membranes, inhibiting respiratory metabolism, hindering the normal function of nucleic acids, and reducing the pathogenicity by inhibiting the transcription level of its main virulence genes [65]. Ju et al. [79] pointed out that the synergistic inhibitory mechanism of eugenol and citral against *Penicillium roqueforti* was to destroy the cell membrane of mycelia; affect the conformation of membrane protein and the fatty acid composition of the cell membrane; and inhibit the enzyme activity, energy metabolism, and gene expression of key molecules in the tricarboxylic acid cycle (TCA) pathway, which led to the disorder of respiration and energy metabolism of *Penicillium roqueforti*, and apoptosis in the end.

*Litsea cubeba* essential oil could also be prepared for coating materials with chitosan, which helped to reduce the consumption of nutrients and delay the decay of fruits such as kumquat and strawberry. Thielmann et al. [80] coated food packaging films with polyvinyl acetate (PVA), containing different concentrations of citral or *Litsea cubeba* essential oil, which showed obvious bactericidal effects on *Escherichia coli* and *Staphylococcus aureus* when dry matter (DM) was standardized to 20% *w*/*w* as the sum of PVA and antimicrobial compound. The headspace packaging for the 20% DM coating also had an inhibitory effect on the growth of yeast and mold in the strawberry application trial.

#### 3.2.2. Antioxidant Activity

*Litsea cubeba* essential oil could be used as a good natural antioxidant in food. However, it is highly volatile and unstable in light or upon heating. Hwang et al. [81] evaluated the DPPH free radical scavenging ability, H_2_O_2_ scavenging activity, and inhibition activity of lipid peroxidation of *Litsea cubeba* essential oil, which proved the significant in vitro antioxidant activity *Litsea cubeba* essential oil. Gogoi et al. [67] reported that methyl heptanone (30.9%) and sabinene (25.22%) were the main components in fruit and leaf essential oils of *Litsea cubeba* Pers. from Northeast India, respectively. Both essential oils showed strong concentration-dependent antioxidant activities, where *Litsea cubeba* fruit essential oil had better antioxidant and anti-inflammatory activities than that from leaves. She et al. [82] studied chemical composition, antimicrobial, and antioxidant activities of *Litsea cubeba* essential oils collected in different months, which found that the yield of *Litsea cubeba* essential oils collected in June, July, and August was 3.47% containing 13 components, 5.02% containing 17 components, and 5.64% containing 17 components, respectively. Neral and geranial accounting for 54.76% were the main components. The essential oil from July had the highest OH· scavenging activity while that from August had the highest DPPH· scavenging activity and ferric-reducing antioxidant activity.

Although *Litsea cubeba* essential oil has been proven to have a considerable antioxidant activity, its particular characteristics like low water solubility, high volatility, and photosensitivity limit its application range. Wang et al. [83] used ultrasonic emulsification to formulate a highly stable and water-dispersible nanoemulsion of *Litsea cubeba* essential oil with improved biological activities like antibiofilm, antioxidant, and antimicrobial properties.

#### 3.2.3. Anthelmintic Activity

Many components in *Litsea cubeba* essential oil such as citral, aromatic alcohol, β-pinene, and citronellal have repelling or lethal effects against mosquitoes. Chen et al. [84] found that *Litsea cubeba* essential oil had a good repellent activity and larvicidal activity against *Aedes albopictus*, which are closely related to its chemical composition. Therefore, the *Litsea cubeba* essential oil may be used for alternative natural repellent to the synthetic DEET.

The insecticidal activity of essential oils from 26 plant species was compared using a fumigation bioassay, where *Litsea cubeba* essential oil was observed with strong insecticidal activity against *Reticulitermes speratus* Kolbe [85]. When *Litsea cubeba* essential oil is used as insecticide in the field or granary, it can kill many kinds of grain pests, such as rice weevil, grain weevil, bean weevil, gnawing beetle, *Rhizopertha dominica*, and so on. Repellent effects of *Litsea cubeba* essential oil on various growth forms of mosquitos and insects are showed in Table 4. 

Wu et al. [87] found that the optimal formula of the repellent liquid was the blending of *Litsea cubeba* oil (10%), *Mentha arvensis* oil (10%), Tween 80 (5%), vanillin (5%), and absolute ethyl alcohol (70%), in which effective sterilizing time for hand skin was 4 h. The blending of *Litsea cubeba* and clove essential oil (40:60 *v*/*v*) had the strongest fumigation activity against female adults of *Liposcelis entomophila* (Enderlein), which showed significant synergism with the co-toxicity factor of 49.42 [90].

#### 3.2.4. Other Bioactivities

Apart from the above, other bioactivities of *Litsea cubeba* essential oil were studied as well. The inhibitory effect of *Litsea cubeba* essential oil on dendritic cell activation was studied, as well as the in vivo immunosuppressive effects in mice [95]. *Litsea cubeba* essential oil was found to inhibit both the production of TNF-α and cytokine IL-12 in lipopolysaccharide-stimulated dendritic cells with a dose-dependent manner, and the hypersensitivity reaction and T-cell infiltration in mice, indicating that *Litsea cubeba* essential oil can potentially be applied in the treatment of contact hypersensitivity, inflammatory diseases, and autoimmune diseases.

Citral, as the major component in *Litsea cubeba* essential oil, could significantly prolong the incubation period of asthma induced by choline phosphate histamine spray, which had certain antiasthmatic, antitussive, and expectorant effects [96]. Furthermore, *Litsea cubeba* essential oil could significantly inhibit the pain response caused by acetic acid and high fever with an analgesic rate of more than 30% [97]. It also had anti-inflammatory and anti-cancer activities [98], which might be used as a new plant resource to prevent such diseases. Gogoi et al. [67] found that essential oils from *Litsea cubeba* fruit and leaves had strong in vitro anti-inflammatory activities, which were relatively low compared to the standard anti-inflammatory drug diclofenac sodium. Choi et al. [99] found that *Litsea cubeba* essential oil could inhibit the production of inflammation mediators (i.e., NO and PGE_2_) in RAW264.7 macrophages activated by lipopolysaccharide, and there was no obvious cytotoxicity when the concentration was lower than 0.01 mg/mL. Similarly, Zhong et al. [100] found that the NO inhibition rate was ≥50% when the concentration of *Litsea cubeba* essential oil was 31.25–62.50 µg/mL through using mouse macrophage cells (RAW264.7) and murine hepatoma cells as models. Quinone reductase could inactivate carcinogens through stress reaction, thus the quinone reductase activity was evaluated as an anti-cancer index, which was induced to double when the concentration of *Litsea cubeba* essential oil was 15.63–31.25 µg/mL.

The mainstream of future beauty market should be natural, safe, and functional. Huang [101] compared the *in vitro* tyrosinase inhibition activity of three volatile oils (*Litsea cubeba*, citronella and clove) and their main components, which showed that *Litsea cubeba* essential oil and its main component citral had a good performance with an inhibition rate of 50%, which showed its development potential in the field of cosmetics and skin care. Furthermore, *Litsea cubeba* essential oil could prolong pentobarbitone-induced mouse sleeping time and had anxiolytic activity and potent analgesic activity [102]. Besides, the inhalation of *Litsea cubeba* essential oil could significantly improve the total mood disturbance and reduce the confusion among the healthy human subjects and the salivary cortisol level at a notable level [103]. This indicated that *Litsea cubeba* essential oil could be used in aromatherapy as a complementary and alternative therapy for health, which is favorable to improve physical, mental, and emotional states [104].

### 3.3. Purification of Litsea cubeba Essential Oil and Its Derivatives

Citral is the main biologically active component in *Litsea cubeba* essential oils, which has two configurations including geranial (α-citral) and neral (β-citral). It is a light yellow volatile oily liquid with a lemon flavor, which is authorized as a food-grade flavor in China. Citral is mainly used in the preparation of lemon, citrus and assorted fruit-type food flavors, dishwashing detergents, soaps, toilet water, and flavoring agents. Also, it can be an intermediate for formulating and synthesizing high value-added functional ingredients, which are often used in perfumes and nutraceuticals.

#### 3.3.1. Purification of *Litsea cubeba* Essential Oil

Citral can be purified from *Litsea cubeba* essential oil in many ways. The traditional steam distillation method has been gradually phased out due to its low utilization rate of raw material and the low purity of the end-product obtained. Conventional chemical method with the addition of sodium sulfite has many deficiencies such as low extraction rate, long reaction time, and the lye contamination after reaction. Liu et al. [105] reduced the original chemical reaction time 20 times with the help of microwave heating, in which temperature and other conditions were flexible and easy to control.

As shown in Table 5, Peng et al. [106] and Fu et al. [107] investigated the effect of phase transfer catalyst type and dosage, reaction time, and temperature on the yield and purity of citral, where the final yield of citral under optimal conditions was 73.47% with the purity of 85.49%. Atmospheric distillation has a high evaporation temperature, and substances with high boiling points are prone to reactions such as decomposition, oxidation, and polymerization. Therefore, vacuum distillation could reduce the boiling point of components in *Litsea cubeba* essential oil, where the reaction proceeded quickly under high vacuum and low temperature without any negative effects on yield, aroma, and quality [108]. Molecular distillation conditions were optimized to improve the yield and purity of citral from *Litsea cubeba* essential oil [109,110]. This technology largely maintains the naturalness of the raw materials, which is an efficient process for industrial applications.

#### 3.3.2. Citral-Derived Fragrances

There are carbonyl groups and double bonds in the structure of citral, which are prone to chemical reactions such as addition, condensation, cyclization, and oxidation. In other words, citral can be a natural synthon for a variety of important perfume compounds with a purity as high as above 90% (Figure 2); this synthetic technique is of great significance to the future development and utilization of *Litsea cubeba* resources.

Ionone is an oily liquid with color ranging from colorless to slightly yellow and a strong violet floral aroma. It is an indispensable raw material or intermediate in the aromatization, in which synthesized products can be divided into α-ionone, β-ionone, and uncommon γ-ionone. α-ionone is often used for the formulation of perfume, and β-ionone can be used for the synthesis of vitamin A. Citral was used to undergo aldol condensation with acetone under the catalysis of NaOH to form pseudoionone, which was then cyclized to produce ionone catalyzed by phosphoric acid with a total yield of 81.4% [111]. Pseudoionone could also be synthesized by solid base (LiOH·H_2_O) catalysis, in which the conversion rate of ciral and the selectivity to pseudoionone under optimal reaction conditions were 98.9% and 97.7%, respectively [112]. Lemonile has a strong and persistent lemon aroma with a higher thermal stability than citral. Citral purified from *Litsea cubeba* essential oils was reacted with hydroxylamine sulfate to form citrate oxime. Under the microwave irradiation, Citral could directly produce lemonile with an average yield of 90.7% using polyethylene glycol 600 as the phase transfer catalyst [113].

Irone is a high-value product used in perfumes, cold cream and cosmetics, cigarettes, soap, and other daily necessities, which has a similar soft sweet smell to iris and violet. The chemical synthesis of irone is mainly accomplished through the citral-pseudoionone pathway. In dilute alkali solution, citral and acetone are condensed to form pseudoionone, which is synthesized by methyl localization, and finally by cyclization. Liu et al. [114] made a break-through in the synthesis of irone, where pseudoirone was firstly formed from methylated pseudoionone, and then cyclized to obtain irone, resulting in a total yield up to 47.9% but with no specific methyl localizer and cyclizing agent reported.

Vitamin A is indispensable for maintaining the normal metabolism of human body, but the extraction process from animal tissues is complicated and costly. Therefore, it is usually chemically synthesized as vitamin A carboxylates mainly through two routes. Roche in Switzerland took β-ionone as the starting material and completed the synthesis of vitamin A acetate through Darzens reaction, Grignard reaction, selective hydrogenation, hydroxyl bromination, and dehydrobromination. BASF in Germany also used β-ionone as the starting material and performed Grignard reaction with acetylene to produce acetylene-β-ionol, and then selectively hydrogenated to obtain ethylene-β-ionol. After Wittng reaction, ethylene-β-ionol condensed with C5 aldehyde to generate vitamin A acetate, which was catalyzed by sodium alkoxide.

## 4. Biorefinery after *Litsea cubeba* Essential Oil Production

*Litsea cubeba* essential oil processing generates numerous residues like kernels and pomaces, which are mostly discarded or burned as wastes, resulting in high environmental burden. Hence, how to transform these wastes to value-added by-products in a sustainable way is the key to open the gate of *Litsea cubeba* industrial market.

### 4.1. Litsea cubeba Kernel Oil

Kernels after the extraction of *Litsea cubeba* essential oil still contain around 40% oil content with high saturation level (≈74%), consisting of mainly medium chain fatty acids like lauric acid and capric acid. The *Litsea cubeba* kernel oil seems to be a promising new source of lauric acid, whose physicochemical properties are comparable to that of coconut oil [115].

As shown in Table 6 [116,117,118,119,120,121], the traditional mechanical method had a low yield with a long pressing time, in which residues still contained a certain amount of oil. The organic solvent extraction is the common industrial method, in which the oil yield could be improved with a lighter color. Aqueous extraction is a traditional method unique to China, whose principle is to separate oil and hydrophilic non-oil components (e.g., protein) based on their differences in affinity to oil and water and specific gravity. This method is easy to operate and can be enhanced with the assistance of microwave or other innovative techniques, which require less equipment with no worry about organic solvent residue problems. The effect of different solvents on the extraction of *Litsea cubeba* kernel oil demonstrated that cyclopentyl methyl ether could be the best substitute to *n*-hexane with comparable extraction efficiency and selectivity [121]. Supercritical CO_2_ extraction as a green extraction technology was also conducted to optimize the influencing parameters, which showed a high extraction efficiency. However, the high initial cost and professional requirements should be taken into consideration for a start-up industrialization.

#### 4.1.1. Purification of Fatty Acids and Derived Green Surfactants

*Litsea cubeba* kernel oil contains more saturated fatty acids than unsaturated fatty acids, where lauric acid content is the highest accounting for about 40–60% of total fatty acids, followed by decanoic acid accounting for about 10–12%. Generally, distillation and freeze-pressing are used to prepare lauric acid, which are difficult to scale up due to high cost and poor quality of products. Zhou et al. [122] refined the *Litsea cubeba* kernel oil through acid refining, saponification, and acidification hydrolysis to obtain mixed fatty acids, where high-purity lauric acid (>98%) was then prepared under vacuum and optimal mixed solvent crystallization. Apart from the medium chain saturated fatty acids (≈72%), there are also mono- (≈22%) and poly- (≈6%) unsaturated fatty acids, which are quite different from other lauric oils [120]. Significantly, a considerable content of lauroleic acid (≈7%) was firstly found in *Litsea cubeba* kernel oil in addition to oleic acid (≈12%) and linoleic acid (≈6%). This special fatty acid may deserve more studies to find its characteristics and functionality.

Surfactant refers to a fine chemical with hydrophilic (i.e., hydroxyl, sulfate, sulfonate) and lipophilic or atmophile non-polar (i.e., hydrocarbon group) groups that can significantly change the liquid surface tension or interfacial tension between two phases. Natural fatty acids in *Litsea cubeba* kernel oil could be purified as a chemical basis for the synthesis of anionic, cationic, and nonionic surfactants. Lauric acid as the main medium chain fatty acid from *Litsea cubeba* kernel oil could be a green synthon for various eco-friendly surfactants such as carboxylate, lauronitrile dodecane nitrile, alkanolamide, *n*-laureylethylenediamine triacetic acid, etc. [123]. Lauryl aldehyde and laurinol derived from *Litsea cubeba* kernel oil could react together to synthesize cyclic acetal type surfactants. Compared to traditional surfactants, this new type of surfactant greatly improved both chemical and biological degradability, as well as surface activity [124].

#### 4.1.2. Biodiesel Production from Kernel Oil

Since the world as an integral whole faces common challenges like depletion of fossil resources, global warming, and an increasing population, plant-based chemicals have gradually become the mainstream toward a petroleum-free sustainable future [125]. Biodiesel is a kind of green renewable energy that has been studied for years.

*Litsea cubeba* kernel oil with a considerable oil content is theoretically feasible to be the biodiesel feedstock. As a non-food vegetable oil, using *Litsea cubeba* kernel oil to produce biofuels conforms to the concept of Green Chemistry without any competition to foods. Wang et al. [126] used the hydrothermal liquefaction to partially convert into bio-oil with a yield of up to 56.9 wt.%, where temperature was found as the most important factor to the final yield. Moreover, Na_2_CO_3_ loading was beneficial to the conversion of the feedstock but adverse to the polymerization pathway of bio-oil formation. Zhang et al. [127] used *Litsea cubeba* kernel oil and ethanol to produce biodiesel with solid acid catalyst SO_4_^2−^/Fe_2_O_3_-TiO_2_. Such a catalyst could improve the reaction activity with the biodiesel yield of 44.9% under the optimal conditions, i.e., molar ratio of ethanol to *Litsea cubeba* kernel oil (16:1), catalyst dosage (10 wt%), reaction time (8 h), and ethanol refluxing temperature (78 °C). Recently, zinc supported on zirconia heterogeneous catalyst (7% Zn/ZrO_2_) was successfully synthesized and performed the best for the biodiesel production from *Litsea cubeba* kernel oil through transesterification under optimal conditions, i.e., methanol to oil molar ratio (8:1), catalyst loading (5 wt%), reaction temperature (120 °C), reaction time (12 h) and stirring rate (400 rpm) [128]. 

#### 4.1.3. Biolubricant Base Oil

Traditional lubricating oils derived from non-renewable mineral raw materials are poorly biodegradable, which are likely to have a serious impact on ecological balance. In contrast, plant oil-based lubricants can be biodegraded quickly and completely with low ecotoxicity [129]. Therefore, the development of ecofriendly biolubricating oils has attracted more attention. According to the ASTM (American Society for Testing and Materials) test, plant oils from rapeseed, soybean, sunflower, and palm with long-chain fatty acids can be used as bio-based raw materials, which can ensure good lubricity and low environmental impact. Nevertheless, using these oils as feedstock may lead to competition with food supply and poor oxidative stability of biolubricant base oils. Recently, *Litsea cubeba* kernel oil demonstrated itself as a promising medium-chain saturated fatty acid feedstock for biolubricant-based oil synthesis [130]. The fatty acid methyl ester (FAME) of *Litsea cubeba* kernel was prepared through decolorization, glycolysis, and methanolysis, which was then transesterified with trimethylolpropane (TMP) to produce trimethylolpropane fatty acid trimester (TFATE) under optimal conditions (Figure 3). The purified TFATE product could achieve good oxidative stability and low-temperature performance qualified by ISO VG 32 standard.

### 4.2. Litsea cubeba Protein

After oil expression, the considerable protein content in the oil cake is often a matter of concern. The extraction rate of *Litsea cubeba* protein reached 31.03% under optimal conditions (i.e., solid–liquid ratio 1:1.5, pH value 11, temperature 35 °C, and time 60 min), where the protein content in the crude protein extract was 75% [131]. The solubility, foaming ability, and foam stability were lower at acidic pH condition. The water holding ability was higher than that of commercial soybean protein isolate when the temperature was below 60 °C.

### 4.3. Litsea cubeba Pomace

*Litsea cubeba* pomace as a waste residue has the potential to be a feed resource due to its comparable nutrient contents to other common fodders, and the trace amount of residual essential oil, whose biological activities may beneficial to the animal growth. Zhang et al. [132] studied metabolic energy and the utilization rate of common nutrients and amino acids on *Litsea cubeba* pomace in feeding roosters and ducks, which showed that the apparent utilization rate of crude *Litsea cubeba* protein and fat was >50% and >80%, respectively. Compared to roosters, ducks had a higher utilization rate of protein and amino acids but with a lower utilization rate of crude fat and fiber, indicating the feeding value of *Litsea cubeba* pomace as poultry fodder. Luo et al. [133] also explored the utilization value of *Litsea cubeba* pomace in ruminant feed compared to *Leymus chinensis* as the control. The rich nutritional value of *Litsea cubeba* pomace was found to be a kind of feed resource. Nonetheless, the inhibitory effect on the growth of rumen microorganisms was found after *Litsea cubeba* pomace digestion, which is unfavorable to rumen fermentation and gas production in ruminants.

In addition, roots and barks in *Litsea cubeba* residual also contain active ingredients such as lignans and alkaloids. A natural dibenzylbutane lignin called 9′-O-di-(E)-feruloyl-meso-5,5′-dimethoxy-secoisolariciresinol (FCL) was found to have anti-inflammatory activity in LPS-induced RAW264.7 cells through the NF-κB signal pathway [134]. Its anti-rheumatoid arthritis and anti-osteoporosis properties were also evaluated by measuring cell proliferation, alkaline phosphatase activity, and calcium deposition formation on osteoblasts [135]. The results revealed that FCL promoted osteoblastgenesis and bone formation, while osteoclastogenesis was suppressed. Importantly, FCL showed strong binding activities to the cathepsin K and mitogen-activated proteinkinase kinase 1 (MEK1), which indicated that FCL may possess anti-osteoporosis potentials targeting cathepsin K and MEK1. Zhang et al. [136] separated five novel isoquinoline alkaloids and one known compound from *Litsea cubeba* and evaluated their in vitro antibacterial, antifungal, and cytotoxic properties, where three alkaloids showed antimicrobial activity against the *Staphylococcus aureus*, *Alternaria alternata*, and *Colletotrichum nicotianae*, while two of them exhibited more potent cytotoxicity against all six tumor cell lines tested.

## 5. Conclusions & Perspectives

In summary, *Litsea cubeba*, as a new woody plant resource, could be comprehensively utilized toward a zero-waste biorefinery with both economic, social, and ecological benefits. Besides its essential oils being the main product (>4000 tons in China) with various biological activities, its residues (>200,000 tons in China) including kernel, protein, and pomace also showed great potential to be further develop as value-added products in different fields. Unlike other woody plant oils used mainly for food, the generic term “*Litsea cubeba* oil” can be used as a food flavoring or be a natural synthon for perfume industry, or even as the feedstock for biofuel productions. The total valorization of *Litsea cubeba* set a successful example for other interesting woody plant resources in China, which may inspire a sustainable biorefinery way for alleviating poverty in underdeveloped or developing countries of origin.

Regarding the current “take-make-dispose” linear economic model, developing a circular economy based on the principles of designing out waste and pollution, keeping products and materials in use, and regenerating natural systems seems to be the only solution to the resource-security problem worldwide. For future development of woody oil industry, recirculation of waste materials should be promoted as a priority through adopting well-targeted policies, legislation, financial, and technical measures. For this, life cycle assessment for products from each woody oil plant resource have to be further made to provide decision makers and the market with quantitative environmental performance information in different phases of the product life cycle, resulting in a scientific foundation from a “cradle to grave” perspective for the sustainable development of *Litsea cubeba* or other plant resources in the same situation.

## Figures and Tables

**Figure 1 molecules-26-03948-f001:**
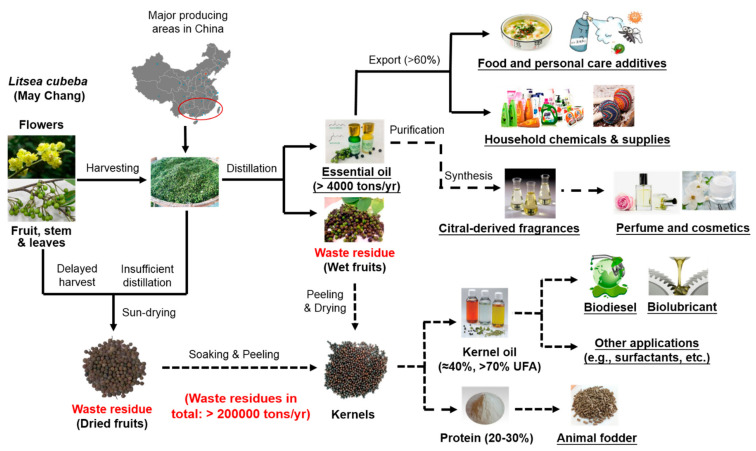
Sustainable biorefinery to produce plant-based products from *Litsea cubeba* (solid line: high technical maturity, half dashed line: medium technical maturity, full dashed line: low technical maturity).

**Figure 2 molecules-26-03948-f002:**
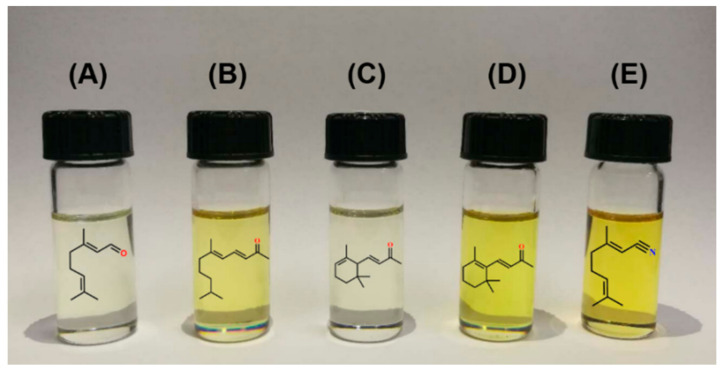
Citral-derived fragrances from *Litsea cubeba* (LC) essential oils: (**A**) purified LC essential oils; (**B**) pseudoionone; (**C**) α-ionone; (**D**) β-ionone; (**E**) lemonile.

**Figure 3 molecules-26-03948-f003:**
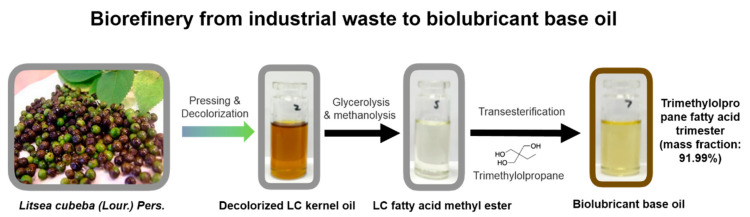
Sustainable biorefinery route for the production of biolubricant base oils from *Litsea cubeba* (LC) kernels as waste residues after the essential oil production.

**Table 1 molecules-26-03948-t001:** Overview of six unique oil crops in China.

Woody Oil Plants	Family	Chemical Composition of Seed Oil	References
Seed Oil Content (%)	Total Unsaturated Fatty Acids (%)	Unique Bioactive Substances	Other Bioactive Substances
*Camellia oleifera*	** 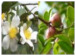 **	Theaceae	1–58	85–93	Camellin	Squalene, tea saponin, lignans, etc.	[5,6]
*Juglans regia*	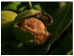	Juglandaceae	59–68	90–91	Juglone	Minerals, melatonin, phospholipid, carotene, vitamin, etc.	[7,8]
*Paeonia suffruticosa*	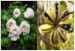	Paeonia	28–31	88–93	Paeonol, paeonoside	Monoterpenes, triterpenes, etc.	[9,10,11]
*Xanthoceras sorbifolium*	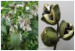	Sapindaceae	55–70	83–88	Nervonic acid, saponin	Sterols, phospholipids, etc.	[12,13]
*Acer truncatum*	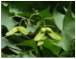	Aceraceae	38–55	90–92	Nervonic acid	Vitamin E, flavonoids, coumarin, etc.	[14,15]
*Eucommia ulmoides*	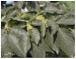	Eucommia	32–37	90–91	Aucubin, chlorogenic acid	Iridoids, phenylpropanoids, lignans, etc.	[16,17]
*Swida wilsoniana*	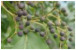	Cornaceae	55–62	74–82	Octacosanol	β-sitosterol, vitamin, etc.	[18,19,20]

**Table 2 molecules-26-03948-t002:** Methods for extracting *Litsea cubeba* essential oils.

Methods	Conditions	Experimental Remarks	References
Microwave-assisted extraction	Microwave irradiation time: 10–20 min; Microwave temperature: 60–80 °C; Distillation temperature: 80–100 °C; Distillation time: 90–120 min.	Compared to conventional steam distillation, the yield of essential oil increased by 36.5–37.5% and the treatment time was four times shorter. The citral content was 5% higher and the amount of its loss in purification reduced by 33.3%	[55]
	Microwave power: 650 W; Extraction time: 40 min; Solid-liquid ratio: 1:4 (g/mL).	Average yield of essential oil was up to 10.29% (g/g).	[56]
Ultrasonic-assisted extraction	Liquid to solid ratio: 3.2:1; Vacuum distillation temperature: 76 °C; Particle size: 80 mesh; Ultrasonic time: 35 min.	The extraction rate under vacuum was 6.94%, which was 33.98% higher than the that of conventional steam distillation. The content of citral was 87.65%.	[57]
	Liquid to solid ratio: 5:1; Ultrasonic time: 25 min; Ultrasonic temperature: 100 °C.	Compared to hydrodistillation and steam distillation, essential oil yield increased with the ultrasonic time; the optimal ultrasonic-assisted extraction was helpful for obtaining high-purity citral.	[58]
Enzymatic-assisted extraction	Heterologous expressed expansin: 500 mL; Cellulase: 2.5 g; Enzymolysis time: 24 h; Temperature: 42 °C; Centrifuge speed: 3000 r/min.	Enzymatic-assisted extraction could significantly improve the yield of essential oil compared to conventional extraction. The highest yield was obtained using composite enzyme (cellulase and expansin), which was 1/3 higher than that using cellulose alone.	[59]
Supercritical CO_2_ extraction	Particle size: 60–80 mesh; Extraction pressure: 25 MPa; Extraction temperature: 45 °C; Extraction time: 60 min; CO_2_ flow rate: 1.5 mL/min.	The extraction rate was up to above 30.19% and the essential oil had clear color.	[60]
Combined extraction	Microwave power: 600 W; Extraction time: 8 min; Solid-liquid ratio: 1:7 (g/mL); Extraction temperature: 85 °C.	Average yield of essential oil of combined extraction assisted by microwave and ultrasound was up to 14.19% (g/g), which was 3.9% higher than that extracted by microwave solely.	[56]
	Oxalic acid/choline chloride: 1:1; Water content: 50%; Liquid-solid ratio: 12:5:1 mL/g Homogenate time: 2 min Microwave power: 700 W	Deep eutectic solvent-homogenate based microwave-assisted hydrodistillation was developed to have a quite different major compounds (e.g., m-cymeme, trans-linalool, etc.) under optimal conditions, which showed higher in vitro radical scavenging activity but lower antifungal activity.	[61]

**Table 3 molecules-26-03948-t003:** Minimal inhibitory concentrations (MICs) of *Litsea cubeba* essential oil against different bacteria and fungi.

Organism		MIC [µg/mL]	Method	References
Bacteria	Gram-Type			
Enterohemorrhagic *Escherichia coli (EHEC)*	−	500	Double dilution	[65]
Methicillin-resistant *Staphylococcus aureus (MRSA)*	+	500	Broth micro-dilution	[66]
*Staphylococcus aureus*	+	80	Broth micro-dilution	[67]
*Bacillus cereus*	+	40	Broth micro-dilution	
*Bacillus subtilis*	+	40	Broth micro-dilution	
*Salmonella typhimurium*	−	20	Broth micro-dilution	
*Listeria monocytogenes*	+	2500	Broth micro-dilution	[49]
*Salmonella*	−	625	Double dilution	[68]
*Shigella*	−	625	Double dilution	
*Pseudomonas aeruginosa*	−	620	Broth micro-dilution	[69]
*Enterococcus faecalis*	+	600	Broth micro-dilution	
*Shewanella putrefaciens*	−	0.5	Broth micro-dilution	[70]
*Staphylococcus albus*	+	11.88–23.75	Broth-dilution	[71]
*Vibrio parahaemolyticus*	−	750	Broth-dilution	[72]
**Fungi**				
*Alternaria alternaria*		0.05	Agar-dilution	[73]
*Aspergillus flavus*		0.5	Agar-dilution	[74]
*Aspergillus niger*		5	Agar-dilution	[73]
*Fusarium oxysporum*		0.49	Agar-dilution	[75]
*Fusarium moniliforme*		0.5	Agar-dilution	[73]
*Fusarium solani*		0.5	Agar-dilution	
*Galactomyces candidum*		1.0	Agar-dilution	[76]
*Candida albicans*		700	Broth micro-dilution	[69]
*Lactobacillus plantarum*		1500	Broth-dilution	[72]
*Malassezia furfur*		2367.61 ± 688.29	Broth micro-dilution	[77]

**Table 4 molecules-26-03948-t004:** Repellent effects of *Litsea cubeba* essential oil on various growth forms of mosquitoes and insects.

Species	Morphology	Experimental Remarks	References
*Aedes aegypti* (L.) mosquitoes	Adult	24 h direct contact mortality: 2.3–20.4% 24 h non-contact mortality: 0–14.3%	[86]
*Aedes albopictus*	Larva (the fourth-instar)	24 h LC_50_: 82.48 µg/mL	[84]
	Pupae	24 h LC_50_: 122.92 µg/mL	
	Adult	73.94 percentage repellency at 20 min (2.0 µL)	[87]
*Lasioderma serricorne*	Adult	Contact toxicity 24 h LC_50_: 27.33 µg/adult Fumigant toxicity 24 h LC_50_: 22.97 mg/L	[88]
*Liposcelis bostrychophila*	Adult	Contact toxicity 24 h LC_50_:71.56 µg/cm^2^ Fumigant toxicity 24 h LC_50_: 0.73 mg/L	
*Bursaphelenchus Xylophilus*	Adult	24 h LC_50_: 0.504 mg/mL	[89]
*Liposcelis entomophila* Enderlein	Adult	24 h LC_50_: 6.23 µL/L	[90]
*Tribolium castaneum*	Adult	1.5 g/cm^2^ repellent rate (12 h): 81.26%	[91]
*Trichoplusia ni*	Larva (the third-instar)	24 h LC_50_: 112.5 µg/larva	[92]
*Luciaphorus perniciosus*	Adult	Contact toxicity 12 h LC_50_: 0.932 µg/cm^2^, 99 µg/cm^2^ showed the highest toxicity causing 97.5 ± 4.1% mortality at 12 h Fumigation 12 h LC_50_: 0.166 µg/cm^3^	[93]
*Anopheles stephensi*	Adult	The protection period: 480 min; 100 percentage repellency	[94]
*Culex quinquefasciatus*	Adult	The protection period: 480 min; 100 percentage repellency	

**Table 5 molecules-26-03948-t005:** Purification technologies of citral.

Methods	Conditions	Experimental Remarks	References
Sodium sulfite chemical addition method	DMSO as phase transfer catalyst: 5% of citral material; Time: 3.5 h, Temperature: 10 °C.	The citral yield was 73.47% and the purity was 85.49%.	[106]
	Methylated-β-cyclodextrin (RM-β-CD) as phase transfer catalyst: 0.65% of citral material; Time: 3.3 h, Temperature: 15 °C.	The citral yield was 86.6% and the purity was 96.5%.	[107]
Vacuum distillation	Pretreatment: dehydration, magnetization, filtration, and deoxidation; Vacuum degree: 10 mm Hg, tower kettle temperature: <100 °C; The first fractionation column top temperature <65 °C, reflux ratio 2:1; The second fractionation column top temperature <90 °C, reflux ratio 3:1.	The purity of citral was 97.9%, and the yield was 90.8%.	[108]
Molecular distillation	Film scraping speed: 400 r/min; Feeding amount: 1 L Cooling water temperature: 12 °C; Distillation temperature: 55 °C; Distillation pressure: 0.18 kPa; Material flow: 15 mL/min.	The purity of citral was up to 98% and the yield rate was up to 77.2%.	[109]
	Film scraping speed: 370–390 r/min; Cooling water temperature: 4–5 °C; Distillation temperature: 45 °C; Distillation pressure: 0.15 kPa; Material flow: 1 drop/s.	The content of citral was increased from 79.61% to 95.08%, and the yield of citral was 80.02%.	[110]

**Table 6 molecules-26-03948-t006:** Extraction methods and technology of *Litsea cubeba* kernel oils.

Methods	Process Conditions	Experimental Remarks	References
Mechanical pressing	Press in a single screw press, collect and filter the crude oil, then store it in a 4 °C refrigerator.	The crude oil yield was 26.2%, which was reduced to 21.2% after simple refining.	[116]
Solvent reflux method	Refluxing with petroleum ether (60–90 °C); Particle size of material: less than 0.15 mm Solid–liquid ratio at 1:14 (g:mL); Extraction temperature and time: 80 °C, 2.0 h.	The yield of oil is 26.69%. The content of lauric acid was 49.53%.	[117]
Microwave-assisted extraction	Microwave time: 65 min; Extraction temperature: 78 °C; Solid–liquid ratio at 1:14.5; Microwave power: 545 W.	The aqueous extraction rate of kernel oil is 29. 36%. The content of lauric acid, capric acid and oleic acid is >50%, 8.512% and 10.603%, respectively.	[118]
	Microwave time: 63 min; Extraction temperature: 69 °C; Solid–liquid ratio at 1:16 (g:mL); Microwave power: 337 W.	The extraction rate of kernel oil is 37.42%, which improved by 30.11% compared to *n*-hexane reflux method. The content of lauric acid is the highest (31.36%).	[119]
Supercritical CO_2_ extraction	Extraction time: 80 min; Extraction temperature: 45 °C; Extraction pressure: 25 MPa; Flow rate of carbon dioxide: 220 L/h.	The extraction rate is above 84.5%, dehulling may increase the extraction rate.	[120]
Alternative solvent extraction	Solid–liquid ratio at 1:20 (g:mL); Heating under reflux for 3 h; Extraction temperature: solvents’ boiling point.	Green solvents were superior to alcoholic solvents with higher oil yields. Alternative solvents to *n*-hexane extracted more micronutrients (e.g, tocopherol, sterol and phenolic compounds) resulting in better antioxidant activities.	[121]

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
