# Peer review of "An Overview on Total Valorization of Litsea cubeba as a New Woody Oil Plant Resource toward a Zero-Waste Biorefinery"

_molecules, 2021, doi:10.3390/molecules26133948_

Round 1

Reviewer 1 Report

The authors deeply analysed the use of Litsea cubeba as plant biorefinery. the work is interesting, readable  and fit the aims of the Journal. 

Moreover I think the paper need a paragraph about LCA or CO2 emission balance to improve the entire work and to speak about circular economy.

I put some suggestions in the attached pdf.

Reviewer 2 Report

The manuscript is interesting and presents a compilation of information contributing to scientific knowledge. It is properly formulated and indicated for publication.

Reviewer 3 Report

The paper makes a review of the valorization of Litsea cubeba and about the extraction methods used as well as the characterization of the principal biological and chemical activities.

The references agree with the review and the tables are well organized. Maybe the tables can be improved, namely where one column begins and the other finished.

The figures are illustrative of the potential sustainable biorefinery for the production of plant-based products from Litsea Cubeba, and discuss.

No special comments to the paper.
